# Predictors of the outcome of immune tolerance induction in patients with haemophilia A and inhibitors: The Brazilian Immune Tolerance (BrazIT) Study protocol

**Ricardo Mesquita Camelo**[1], **Daniel Gonçalves Chaves**[2], **Luciana Werneck Zuccherato**[1,3], **Suely Meireles Rezende**[1]*, **for the BrazIT Study Team**[¶]

1 Faculty of Medicine, Universidade Federal de Minas Gerais, Belo Horizonte, Brazil, 2 Fundação HEMOMINAS, Belo Horizonte, Brazil, 3 Center of Education and Research, Instituto Mario Penna, Belo Horizonte, Brazil

¶ Membership of the BrazIT Study Team is listed in the Acknowledgments.
* srezende@medicina.ufmg.br, suely.rezende@uol.com.br

**Funding:** This study is entirely funded by the Brazilian Government grants from Fundo Nacional de Saúde (Grant number 17217.9850001-15-006)

## Abstract

The development of inhibitors is the main complication of haemophilia A (HA) treatment. Immune tolerance induction (ITI) is the treatment of choice for inhibitor eradication. We describe the methodology of the Brazilian Immune Tolerance Induction (BrazIT) Study, aimed to identify clinical, genetic, and immune biomarkers associated with response to ITI and inhibitor recurrence. This cohort study includes people with HA (PwHA) and inhibitors (a) who require bypassing agents to treat and/or prevent bleeding, and (b) who are at any stage of ITI treatment. Patients are included in each haemophilia treatment centre (HTC). Factor VIII (FVIII) and inhibitor assessments are performed at local laboratories of each HTC. The ITI regimen followed the national protocol of the Brazilian Ministry of Health. All PwHA starts with low-dose ITI (50 IU/kg three times weekly); high-dose regimen (100 IU/kg daily) is used if there is lack of response to the low-dose ITI. Outcomes are classified as total or partial success, and failure. Standardized case report forms with clinical, laboratory, and treatment data are collected from medical files and interviews. Blood samples are collected for genetic and immune biomarkers at the time of inclusion in the study and at the end of ITI. The study is ongoing and, currently, 202/250 (80.8%) PwHA from 15 HTCs have been included. BrazIT Study is the largest cohort of PwHA and inhibitor under treatment with the same ITI regimen reported to date. This study is likely to contribute with novel predictors of ITI response.

## Introduction

Haemophilia A is a rare inherited disorder caused by diminished/absent coagulation factor VIII (FVIII) activity, which result in spontaneous haemorrhages [1]. Intravenous FVIII infusions are usually indicated to treat or prevent bleeding episodes [1]. Unfortunately,

and CNPq (Grant number 420008/2018-7). RMC received a scholarship (PDSE-88881.362041/2019-1) from CAPES, an agency of the Brazilian Ministry of Education, to carry out part of his Philosophiæ Doctorate as a visiting student at Leids Universitair Medisch Centrum, in the Netherlands.

**Competing interests:** RMC has received speaker/consultant fees and scientific event grants from Hoffman-La Roche and Takeda. DGC and LWZ have no conflicts of interest to declare. SMR works as an advisor to the Brazilian Program of Inherited Bleeding Disorders (Brazilian Ministry of Health) and received consultancy fees from the Brazilian Ministry of Health. This does not alter our adherence to all PLOS ONE policies on sharing data and materials. After the publication of the final analysis, the data can be shared upon request and guarantee of the confidentiality of each participant. Furthermore, this must adhere to the rules of the Brazilian ethical resolutions for the development of research with human beings.

**Abbreviations:** EDTA, ethylenediaminetetraacetic acid; ELISA, enzyme-linked immunosorbent assay; FVIII, factor VIII; HTC, Haemophilia Treatment Centre; IgG, immunoglobulin G; ITI, immune tolerance induction; MoH, Ministry of Health; PwHA, person/people with haemophilia A.

neutralizing antibodies against FVIII–inhibitors–can develop in up to 40% of people with haemophilia A (PwHA) [2]. Although inhibitors mainly interferes with FVIII interactions by targeting some of its epitopes, they can also inactivate FVIII by proteolysis [3]. Besides increasing the treatment costs, inhibitors are associated with frequent and difficult-to-treat bleeding, muscle and joint complications, higher mortality, and reduced quality of life [4–8].

Currently, the only treatment capable of eradicating inhibitors is the immune tolerance induction (ITI), which consists of frequent infusions of FVIII to induce anergy [9]. This treatment is long-lasting, expensive, and can be troublesome, due to the need of high adherence, good venous access, adequate medical support, and continuous monitoring [9–11]. Moreover, the success rates among the different protocols worldwide range from as low as 60% to around 90% [12–16].

Little is known about predictors of response to ITI. Consistently recognised predictive factors are mainly patient-related (e.g., inhibitor historic peak titre, inhibitor titre at ITI start, and peak during ITI), but there are controversies [17–21]. For instance, historical inhibitor peak titre has been evaluated in six registries [17, 18, 20–23], but only two reported it as predictive of ITI response [20, 21]. Additionally, other factors whose role needs to be confirmed (e.g., *F8* genotypes, immune biomarkers, age at ITI start, and time elapsed from inhibitor diagnosis and ITI start) were not extensively evaluated [9, 10].

Furthermore, few reports have explored clinical and laboratory variables associated with tolerance maintenance after ITI. Although some physicians have advocated resuming regular and continuous prophylaxis with FVIII to avoid inhibitor recurrence, a retrospective cohort do not support this [24]. Moreover, this study demonstrated that the risk of inhibitor recurrence after successful ITI was associated with a low recovery of FVIII and the use of immunomodulatory therapy [24]. The overall recurrence rate of inhibitors was higher in this study [24] than in the aforementioned registries [19, 21, 25].

Different results may be due to the heterogeneity of the registries, including number of enrolled patients, follow-up periods, diversity of ITI and post-ITI regimens, use of immunosuppressors/immunomodulators, and definitions of outcomes [9–11]. The Brazilian ITI program was implemented as a National Government public policy in 2011 [26]. Moreover, in Brazil, factor concentrates for haemophilia care are purchased centrally by the Ministry of Health (MoH) and distributed to HTCs across the country [27, 28]. Currently, about 450 PwHA have been included in this program. This large population of PwHA on ITI provided us an outstanding opportunity to initiate a cohort study: the Brazilian Immune Tolerance (BrazIT) Study. Therefore, the present study aims to evaluate predictors of ITI response in these patients. In addition, tolerant PwHA are followed for at least one year after finishing ITI to evaluate inhibitor recurrence.

## Methodology

### Setting

The study is conducted at the Faculty of Medicine, Universidade Federal de Minas Gerais, Belo Horizonte, Brazil. Patients are enrolled in 15 Haemophilia Treatment Centres (HTC) distributed over the five Brazilian geographical regions. Candidate PwHAs are invited to participate by the attending physician (member of the interdisciplinary team of the HTC), and the decision takes into consideration the interests of the patient and/or his/her tutor. A written informed consent is signed upon acceptance to participate in the study.

### Patients

Inclusion criteria are PwHA and inhibitors (a) who require the use of bypassing agents for treatment or prevention of bleeding and (b) who are at any stage of ITI treatment, which can

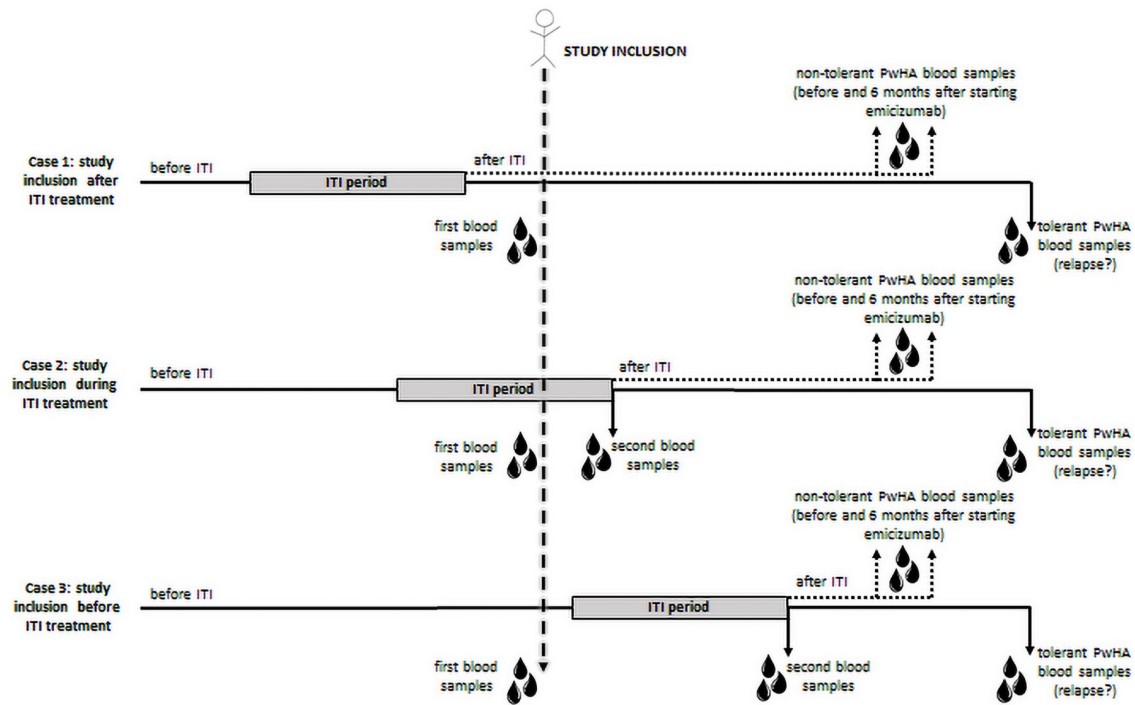

**Fig 1. Illustrative chart of patients' inclusion and collection of samples in the BrazIT Study.** The inclusion of patients occurs before, during or after ITI. Blood samples are collected before and after ITI if patients are included before ITI initiation; during and after ITI if patients are included during ITI; and after ITI if patients are included after ITI. Another blood sample is collected during the follow up, until the end of the study. ITI, immune tolerance induction.

be before starting (Case 1), during (Case 2) or after the end of ITI (Case 3; Fig 1). Patients are included regardless of age, sex, haemophilia severity, or inhibitor titre. Inhibitor is considered positive when titre is $\geq$ 0.6 BU/mL, confirmed by another assessment between at least 2–4 weeks apart. Patients are included in each HTC; FVIII and inhibitor assessments are performed at the local haemostasis laboratories of each HTC.

## Study design and ethical approval

BrazIT Study is a non-interventional cohort study. The study was approved by the ethical committee of Universidade Federal de Minas Gerais (CAAE 52812415.8.0000.5149, 15/Apr/2016), which coordinated the study in the whole country. All included patients or their guardians sign a consent form.

## The Brazilian Immune Tolerance Induction Protocol

Nationally, the ITI regimen follows the Brazilian Program of Inherited Bleeding Disorders of the MoH [26]. The ITI treatment is a decision made between the physician (part of the interdisciplinary team of the HTC) and the patient and/or his/her tutor. Briefly, ITI is performed by using the same type of FVIII concentrate against which inhibitor was developed (e.g., either plasma-derived or recombinant). Low-dose ITI (50 IU/kg three times weekly) is used for all PwHA as the initial ITI regimen. After the first 6 months of ITI, if inhibitor titre does not reduce more than 20% after inhibitor peak, the regimen can be enhanced to 100 IU/kg every day (high-dose regimen). Upon lack of response with high-dose ITI with recombinant FVIII, plasma-derived FVIII can be used as an alternative. Prophylaxis with bypassing agent is

recommended if the patient has a bleeding phenotype; however, when inhibitor titre decreases below 5 BU/mL, it should be suspended [29].

The criteria of ITI outcomes are based on the definitions stated by Hay and DiMichele [30] and are classified as success (i.e., the PwHA responds to exogenous FVIII at regular or higher doses) and failure (i.e., the PwHA does not respond to exogenous FVIII, besides the ITI). Total success is achieved when (a) inhibitor titre is negative at least twice, (b) FVIII pharmacokinetics is normal (half-life is $\geq$ 6 h and recovery is $\geq$ 66%), and (c) usual doses of FVIII are used to treat or prevent bleeding. Partial success is defined as (a) inhibitor titre is 0.6–2.0 BU/mL, and/or (b) FVIII pharmacokinetics is not normal, and/or (c) the patient needs higher-than-usual FVIII doses to treat or prevent bleeding. If the success parameters are not reached after 33 months of ITI, the treatment is considered as failure.

Both FVIII activity assays and inhibitor titre tests are performed in each HTC, after training by the Brazilian Laboratory Committee of Bleeding Disorders. FVIII activity could be evaluated by one-stage or chromogenic assays [31]. Inhibitor titre is measured according to the Nijmegen-modified Bethesda assay [32]. All laboratories have external quality assessment control in place.

### Follow-up

All PwHA who finished ITI with success (total and partial) or failure are followed for at least one year. FVIII and/or inhibitor assessments are performed every 6 months.

### Inhibitor recurrence among tolerant patients

Inhibitor recurrence is defined as any inhibitor titre $\geq$ 0.6 BU/mL on two or more occasions [24].

### Emicizumab prophylaxis

Currently, PwHA and inhibitors who failed ITI are treated with bypassing agents [29]. The Brazilian MoH has approved the use of emicizumab for prophylaxis of PwHA and inhibitors at any age who failed ITI. The product is meant be available by the end of 2021. Loading doses of 3.0 mg/kg weekly, for four weeks, will be administered subcutaneously, followed by maintenance doses of 1.5 mg/kg weekly or 3.0 mg/kg every two weeks [1, 33].

### Patient data

Standardized case report forms with clinical, laboratory, and treatment data are collected from medical files and interviews about different time-points: before, during, and at the end of ITI, and during follow-up until study closure, inhibitor relapse, or emicizumab prophylaxis beginning. The patient data which are collected in the BrazIT Study were depicted on the Table 1.

### Blood analyses

Venous blood samples are collected from all PwHA at the inclusion in the study, which can be before starting (Case 1), during (Case 2) or after the end of ITI (Case 3), and at the end of ITI from PwHA included before or during ITI (Fig 1). For tolerant PwHA whose inhibitor relapses, an additional blood sample is collected at the time of recurrence (Fig 1). For the ones with indication of emicizumab prophylaxis, blood samples are collected before and at 6 months after starting emicizumab (Fig 1).

**Table 1. Clinical and treatment data collected in the BrazIT Study, using a validated case report form.**

| |
|---|
| **Before ITI** |
| HTC, birth date, sex, skin colour, haemophilia diagnosis (date, first and lowest FVIII activity levels), first FVIII infusion (date and reason), inhibitor diagnosis (date and titre), family history of haemophilia and inhibitor, and inhibitor historic peak |
| **During ITI** |
| Dates of ITI start and withdrawal, inhibitor titres immediately before and along the treatment (with dates), venous access type, initial ITI regimen and changes, need of bypassing agent infusion (type and treatment modality), bleeding episodes, and surgical procedures |
| **ITI end** |
| Reason for ITI withdrawal (when this occurred), FVIII activity levels (with dates), inhibitor titres (with dates), duration of ITI, and inhibitor titre and FVIII pharmacokinetics (recovery and half-life), if success |
| **Follow-up after ITI** |
| Dates of start and end of follow-up, treatment modalities and regimens, clotting factor types, inhibitor titres (with dates), bleeding episodes, and surgical procedures |
| • For those who had successful ITI and then relapsed: inhibitor assessment (date and titres) |
| • For those who failed ITI and started emicizumab prophylaxis: emicizumab regimen |

FVIII, factor VIII; HTC, haemophilia treatment centre; ITI, immune tolerance induction.

Venous blood samples are collected in EDTA and citrated tubes, centrifuged for 10 min at 1,500 rpm, and stored at -80˚C until analysis. Blood cells from EDTA tubes are used for genetic tests and plasma samples from citrated tubes are used for assessment of immune biomarkers.

Genetic testing is performed only once. Intron 1 and intron 22 inversions in *F8* are performed according to established protocols [34]. The assessment of variants in genes associated with the disease itself or with the immune response is performed by enrichment panels for targeted whole exome sequencing (xGen Exome Research Panel v2, xGen CNV Backbone Panel and xGen Human mtDNA Research Panel kits, IDT, Newark, USA) spanning 34 Mb and 19,433 genes. Sequencing is carried out on the NovaSeq 6000 (Illumina, San Diego, USA) StrandOmics platform, based on the GRCh37/hg19 version of the human genome, is used for bioinformatics analysis.

Immune biomarkers are evaluated at each time point of blood sampling. Specific anti-FVIII IgG1 and IgG4 are detected using in-house ELISA [35]. Plasma concentrations of chemokines (CXCL8/IL-8, CXCL9/MIG, CXCL10/IP-10, CCL2/MCP-1, CCL5/RANTES) and cytokines (IL-2, IL-4, IL-6, IL-10, IL-17A, IFN-gamma, and TNF) are measured according to the recommendations of the commercial kits (CBA–Cytometric Bead Array; BD Biosciences, San Jose, USA) [35].

## Safety assessments

The study has no therapeutic intervention. The risks involve (a) breach of confidentiality and (b) puncture-site haematomas. To avoid the first, included PwHA are registered on an Excel data sheet as numbers, without their respective names or HTC registries. The Excel file is encrypted, and access is protected by login and password. To avoid haematomas, only healthcare professionals with experience in venous puncture to collect blood samples perform blood sampling. Blood is collected from the same access a clotting factor is infused for treatment. Blood sampling is performed before the clotting factor infusion. After venous puncture, soft-to-moderate compression is applied on the puncture site for 5 to 10 min, until no haemorrhage is observed.

There will be no direct benefit for the participants of this study. The results of this study will potentially benefit PwHA and inhibitor in the future, in case it is successful in determining factors and/or biomarkers (predictors) of ITI success and/or failure. This can be of major impact in ITI treatment as long as it could avoid initiation of ITI in PwHA who might fail and support the indication of ITI in the ones who might respond. For the healthcare system, it will likely avoid the waste of time and resources, which could be better applied somewhere else.

Although the BrazIT Study is a non-interventional study, safety concerns related to medicines, blood products, and treatments are reported according to the haemophilia directives and the Brazilian regulation [36–40].

## Discussion

About 450 PwHA and inhibitors have been treated with the Brazilian ITI Protocol since 2012. We therefore took advantage of this large, admixed population of PwHA on ITI and initiated the BrazIT Study which aims to evaluate predictors of response to ITI. We plan to include 250 PwHA, which is, to our knowledge, the largest cohort of PwHA and inhibitor under ITI reported to date. During the preparation of this manuscript, a total of 202/250 (80.8%) PwHA had already been included in this study. We plan to close the study inclusion in December 2021, and the PwHA will be followed up until 1 year after the end of ITI.

The methodology of the study is unique because it is designed to collect socio-demographic, clinical, and laboratory data in three time-points during PwHA treatment–before the initiation, during, and at the end of ITI–, and during follow-up after ITI has been finished. Blood samples are also collected at these time-points. Blood cells are used for exome sequencing, including *F8* genotyping. Plasma samples are collected to assess immune biomarkers [9, 41]. We are particularly interested on evaluating whether there is any association between specific immune biomarkers at both genotype and phenotype levels and the outcome of ITI. During follow-up of tolerized PwHA, we are also going to evaluate inhibitor recurrence incidence and risk factors. To our knowledge, this approach has not so far been explored.

The first report of ITI was performed in a 16-year-old patient who received cyclophosphamide, prednisone, and large doses of FVIII and responded clinically to the treatment in a few days [42]. More than 20 years later, the Bonn Protocol was published, which consisted of an intensive FVIII treatment (100–150 IU/kg twice daily), in addition to partially activated prothrombin complex concentrate (50 U/kg twice daily), in case of a bleeding phenotype [13, 14]. Seventy-two PwHA and inhibitors were treated resulting in a success rate of 86.7%, defined by negative inhibitor and normal FVIII half-life [13, 14]. Since then, other ITI protocols have been used worldwide, with different regimens and success rates [9, 10]. Finally, the only clinical trial published to date has found no difference in the response rate between two regimens of ITI: a high-dose FVIII (200 IU/kg/day) and a low-dose FVIII (50 IU/kg 3 times/week) regimen [30]. However, PwHA and inhibitors who received high-dose regimen responded faster and experienced less bleeding events [30]. Therefore, the World Federation of Haemophilia recognizes that the optimal regimen for ITI remains to be defined, and no specific regimen is recommended [1].

The risk of recurrence after successful inhibitor eradication affects the cost-effectiveness of ITI. However, maintaining tolerance appears to be a stable condition, with relapse of the inhibitor not being observed frequently. Relapse rates have varied from 3.8% to 12.5% in some small retrospective series with follow-up periods from 5.0 to 8.5 years [15, 19, 20, 43–45]. One large registry reported relapse of 6/128 (4.7%) tolerant PwHA from 1 to 15 years with a cumulative risk of 15% over 15 years [21]. A multicentre retrospective cohort consisting of 64 tolerant PwHA showed that a recurrent inhibitor titre ≥ 0.6 BU/mL occurred at least once in 19

(29.7%) and more than once in 12 (18.8%), resulting in a probability of any recurrent inhibitor at 1 and 5 years of 12.8% and 32.5%, respectively [24]. Inhibitor recurrence was associated with having received immunomodulation during ITI and FVIII recovery of < 85% at the end of ITI, but not with adherence to with prophylactic FVIII infusion after ITI [24].

For PwHA and inhibitors who do not respond to ITI, increased treatment costs, frequent and difficult-to-treat bleeding, muscle and joint complications, higher mortality, and reduced quality of life are still a dilemma [4–8]. Until recently, the only alternative to avoid bleeding in such cases was prophylaxis with bypassing agents, which usually has an effectiveness up to 72% [29]. Recently, a bispecific monoclonal recombinant antibody that mimics the FVIII activity, without the interference of anti-FVIII inhibitors–emicizumab–was introduced in the haemophilia therapeutics pipeline [46]. Emicizumab is used as a prophylactic regimen and effectively replaces the bypassing agents, requiring a more feasible regimen and administration route [47–49]. In Brazil, it was approved for prophylaxis of PwHA and inhibitors who failed ITI. One of the aims of the BrazIT Study is to evaluate the immune biomarkers profile before and after emicizumab therapy.

The BrazIT Study has several strengths. Firstly, this study comprises the largest cohort of PwHA and inhibitors treated with ITI to date. Only one previous registry included more PwHA and inhibitor and this was the International Immune Tolerance Registry which included 314 patients from 50 HTCs of 12 countries [21]. However, the ITI protocols included in this registry were highly heterogeneous, with FVIII regimen ranging from less than 50 IU/kg/day up to more than 200 IU/kg/day, and 5.1% received steroids [21]. In the BrazIT Study, a unique, national ITI protocol is used, with specific brands of FVIII concentrates. Secondly, although several ITI registries have been published to date, some have small sample size and ITI regimen was heterogeneous in the largest ones [9–11]. Therefore, well designed studies with large population of PwHA and inhibitors focusing on the investigation of factors related with response to ITI (prognostic studies) are still missing. This type of study is strategic for the field, once the identification of predictors of non-response could avoid the burden and costs of ITI in non-responsive patients. Based on prediction of ITI failure, some physicians may argue against initiating ITI, favouring prophylaxis with emicizumab [50]. However, despite the advent of emicizumab as an effective agent to prevent bleeding in PwHA and inhibitors, ITI is still recommended as the first line treatment to eradicate inhibitors [1, 33]. Thirdly, the BrazIT Study is aimed to identify new predictive factors related to ITI response, which includes not only clinical characteristics but also immune and genetic (other than *F8*) biomarkers. The involvement of immune modulators in the biogenesis of inhibitors has been described [51, 52], but, to our concern, this is the first study targeting these biomarkers as potential predictors of ITI outcome. Fourthly, since inhibitor status is evaluated regularly among Brazilian PwHA, this study may determine both recurrence incidence and risk factors for relapse in PwHA who responded to ITI. Considering the lowest ITI success rate reported to date (~60%) [43, 53–55], we expect to evaluate recurrence among at least 150 PwHA. Finally, immune biomarkers profiles before and after emicizumab therapy in PwHA who failed ITI can be assessed for the first time in the literature.

## Conclusion

In conclusion, we presented the methodology of the BrazIT Study, which is, to date, the largest cohort of PwHA and inhibitor under ITI using the same protocol. This study is likely to contribute to new data on predictive factors related to ITI response, which could impact on the discovery of potential targets for therapeutics and to individualized treatment.

## Acknowledgments

The authors thank the patients, guardians, and all staff from the Haemophilia Treatment Centres for supporting this study.

The BrazIT Study Group is composed by Dr. Maise Moreira Dias and Dr. Laura Peixoto Magalhaes from FM-UFMG, Dr. Ieda Pinto from HEMOPA, Dr. Daniele Campos Fontes Neves from FHEMERON, Dr. Maria Aline Cerqueira from HEMOPI, Dr. Rosangela Albuquerque Ribeiro from HEMOCE, Dr. Edvis Serafim from HEMONORTE, Dr. Leina Etto from HEMOIBA, Dr. Fabia Michelle Rodrigues de Araujo Callado from HEMOPE, Dr. Andrea Gonçalves de Oliveira from HEMOMINAS, Dr. Monica Hermida Cerqueira from HEMORIO, Dr. Doralice Tan from FAMEMA, Dr. Andrea Aparecida Garcia from Hemocentro de São José do Rio Preto, Dr. Maria do Rosario Ferraz Roberti from HEMOGO, Dr. Claudia Santos Lorenzato from HEMEPAR, Dr. Tania Anegawa from UEL-HEMEPAR, and Dr. Vivian Karla Brognoli Franco from HEMOSC.

## Author Contributions

**Conceptualization:** Suely Meireles Rezende.

**Data curation:** Ricardo Mesquita Camelo, Daniel Gonçalves Chaves, Luciana Werneck Zuccherato.

**Formal analysis:** Ricardo Mesquita Camelo, Daniel Gonçalves Chaves, Luciana Werneck Zuccherato.

**Funding acquisition:** Suely Meireles Rezende.

**Writing – original draft:** Ricardo Mesquita Camelo, Suely Meireles Rezende.

**Writing – review & editing:** Ricardo Mesquita Camelo, Daniel Gonçalves Chaves, Luciana Werneck Zuccherato, Suely Meireles Rezende.

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
