## [Decision Letter · Decision Letter 0]

26 Jul 2021

PONE-D-21-13311

Predictors of the outcome of immune tolerance induction in patients with haemophilia A and inhibitors: the Brazilian Immune Tolerance (BrazIT) Study protocol

PLOS ONE

Dear Dr. Camelo,

Thank you for submitting your manuscript to PLOS ONE. After careful consideration, we feel that it has merit but does not fully meet PLOS ONE’s publication criteria as it currently stands. Therefore, we invite you to submit a revised version of the manuscript that addresses the points raised during the review process.

The manuscript is interesting. Authors should add information about the statistical methods that will be used for calculating the number that will be enrolled and analyzing the final data sets

We look forward to receiving your revised manuscript.

Kind regards,

Dominique Heymann, Ph.D.

Academic Editor

PLOS ONE

Journal Requirements:

RMC has received speaker/consultant fees and scientific event grants from Hoffman-La Roche and Takeda. DGC and LWZ have no conflicts of interest to declare. SMR works as an advisor to the Brazilian Program of Inherited Bleeding Disorders (Brazilian Ministry of Health) and received consultancy fees from the Brazilian Ministry of Health.  

Additional Editor Comments:

Reviewers' comments:

Reviewer's Responses to Questions

**Comments to the Author**

1. Does the manuscript provide a valid rationale for the proposed study, with clearly identified and justified research questions?

Reviewer #1: Yes

2. Is the protocol technically sound and planned in a manner that will lead to a meaningful outcome and allow testing the stated hypotheses?

Reviewer #1: Yes

3. Is the methodology feasible and described in sufficient detail to allow the work to be replicable?

Reviewer #1: Yes

4. Have the authors described where all data underlying the findings will be made available when the study is complete?

Reviewer #1: Yes

5. Is the manuscript presented in an intelligible fashion and written in standard English?

Reviewer #1: Yes

6. Review Comments to the Author

You may also provide optional suggestions and comments to authors that they might find helpful in planning their study.

Reviewer #1: This study protocol is very excellent, well-prepared, and may provide valuable information for management of hemophiliac patient with inhibitor. I really appreciate author's work in this protocol, and I'm eager for the upcoming result of the study.

7. PLOS authors have the option to publish the peer review history of their article (what does this mean?). If published, this will include your full peer review and any attached files.

Reviewer #1: No

---

## [Author Response · Author response to Decision Letter 0]

30 Jul 2021

Rebuttal letter

July 28th, 2021

Dr. Dominique Heymann, Ph.D.

Academic Editor

PLOS ONE

Dear Editor,

On behalf of all co-authors, I thank you and the reviewers for your time and efforts on revising the manuscript. We now submit the revised text.

I hereby provide responses to the suggestions and, when appropriate, the changes we made. The amendments were indicated in track changes in the original file, as recommended. Below, you will find them with our comments highlighted in red.

Concerning potential competing interests, RMC has received speaker/consultant fees and scientific event grants from Hoffman-La Roche and Takeda. DGC and LWZ have no conflicts of interest to declare. SMR works as an advisor to the Brazilian Program of Inherited Bleeding Disorders (Brazilian Ministry of Health) and received consultancy fees from the Brazilian Ministry of Health. However, this does not alter our adherence to PLOS ONE policies on sharing data and materials. After publication of the final analysis, data can be shared upon request and guarantee of the confidentiality of each participant. Furthermore, this must adhere to the rules of the Brazilian ethical resolutions for the development of research with human beings.

This manuscript has not been submitted to another journal.

I hope the manuscript is now suitable for publication.

Best wishes,

Suely M. Rezende.

 

PONE-D-21-13311

Predictors of the outcome of immune tolerance induction in patients with haemophilia A and inhibitors: the Brazilian Immune Tolerance (BrazIT) Study protocol

PLOS ONE

Dear Dr. Camelo,

Thank you for submitting your manuscript to PLOS ONE. After careful consideration, we feel that it has merit but does not fully meet PLOS ONE’s publication criteria as it currently stands. Therefore, we invite you to submit a revised version of the manuscript that addresses the points raised during the review process.

The manuscript is interesting. Authors should add information about the statistical methods that will be used for calculating the number that will be enrolled and analyzing the final data sets

We thank the editor for this comment. A sample size calculation was not performed. However, even before the end of the enrollment, we have already included 210 patients. This is the largest cohort of patients with hemophilia A and inhibitors who underwent to ITI to date. We plan to include up to 250 patients.

• A rebuttal letter that responds to each point raised by the academic editor and reviewer(s). You should upload this letter as a separate file labeled 'Response to Reviewers'. This is the rebuttal letter.

• A marked-up copy of your manuscript that highlights changes made to the original version. You should upload this as a separate file labeled 'Revised Manuscript with Track Changes'. It was uploaded.

• An unmarked version of your revised paper without tracked changes. You should upload this as a separate file labeled 'Manuscript'. It was uploaded.

No changes were made.

If applicable, we recommend that you deposit your laboratory protocols in protocols.io to enhance the reproducibility of your results. Protocols.io assigns your protocol its own identifier (DOI) so that it can be cited independently in the future. For instructions see: http://journals.plos.org/plosone/s/submission-guidelines#loc-laboratory-protocols. Additionally, PLOS ONE offers an option for publishing peer-reviewed Lab Protocol articles, which describe protocols hosted on protocols.io. Read more information on sharing protocols at https://plos.org/protocols?utm_medium=editorial-email&utm_source=authorletters&utm_campaign=protocols. We opted for not depositing our protocols yet.

We look forward to receiving your revised manuscript.

Kind regards,

Dominique Heymann, Ph.D.

Academic Editor

PLOS ONE

Journal Requirements:

https://journals.plos.org/plosone/s/file?id=wjVg/PLOSOne_formatting_sample_main_body.pdf and https://journals.plos.org/plosone/s/file?id=ba62/PLOSOne_formatting_sample_title_authors_affiliations.pdf I formatted the text.

RMC has received speaker/consultant fees and scientific event grants from Hoffman-La Roche and Takeda. DGC and LWZ have no conflicts of interest to declare. SMR works as an advisor to the Brazilian Program of Inherited Bleeding Disorders (Brazilian Ministry of Health) and received consultancy fees from the Brazilian Ministry of Health.

We added two sentences (in italic) to the Competing Interests part:

RMC has received speaker/consultant fees and scientific event grants from Hoffman-La Roche and Takeda. DGC and LWZ have no conflicts of interest to declare. SMR works as an advisor to the Brazilian Program of Inherited Bleeding Disorders (Brazilian Ministry of Health) and received consultancy fees from the Brazilian Ministry of Health. This does not alter our adherence to all PLOS ONE policies on sharing data and materials. After the publication of the final analysis, the data can be shared upon request and guarantee of the confidentiality of each participant. Furthermore, this must adhere to the rules of the Brazilian ethical resolutions for the development of research with human beings.

Please include your updated Competing Interests statement in your cover letter; we will change the online submission form on your behalf. We added the above statement to the cover letter.

3. Your ethics statement should only appear in the Methods section of your manuscript. If your ethics statement is written in any section besides the Methods, please delete it from any other section. We have changed it accordingly.

After considerations, we have decided to delete the Fig. 1, as well as its citations and caption. Then the previous Fig. 2 turned is now Fig. 1, and its citations and caption were changed.

5. Please include captions for your Supporting Information files at the end of your manuscript, and update any in-text citations to match accordingly. Please see our Supporting Information guidelines for more information: http://journals.plos.org/plosone/s/supporting-information. No Supporting Information file was included in our manuscript.

Additional Editor Comments:

Please review your reference list to ensure that it is complete and correct. If you have cited papers that have been retracted, please include the rationale for doing so in the manuscript text, or remove these references and replace them with relevant current references. Any changes to the reference list should be mentioned in the rebuttal letter that accompanies your revised manuscript. If you need to cite a retracted article, indicate the article’s retracted status in the References list and also include a citation and full reference for the retraction notice. No changes were made.

Reviewers' comments:

Reviewer's Responses to Questions

Comments to the Author

1. Does the manuscript provide a valid rationale for the proposed study, with clearly identified and justified research questions?

Reviewer #1: Yes

2. Is the protocol technically sound and planned in a manner that will lead to a meaningful outcome and allow testing the stated hypotheses?

Reviewer #1: Yes

3. Is the methodology feasible and described in sufficient detail to allow the work to be replicable?

Reviewer #1: Yes

4. Have the authors described where all data underlying the findings will be made available when the study is complete?

Reviewer #1: Yes

5. Is the manuscript presented in an intelligible fashion and written in standard English?

Reviewer #1: Yes

6. Review Comments to the Author

You may also provide optional suggestions and comments to authors that they might find helpful in planning their study.

Reviewer #1: This study protocol is very excellent, well-prepared, and may provide valuable information for management of hemophiliac patient with inhibitor. I really appreciate author's work in this protocol, and I'm eager for the upcoming result of the study.

We thank the reviewer for this compliment.

7. PLOS authors have the option to publish the peer review history of their article (what does this mean?). If published, this will include your full peer review and any attached files.

Do you want your identity to be public for this peer review? For information about this choice, including consent withdrawal, please see our Privacy Policy.

Reviewer #1: No

---

## [Editor Report · Decision Letter 1]

4 Aug 2021

Predictors of the outcome of immune tolerance induction in patients with haemophilia A and inhibitors: the Brazilian Immune Tolerance (BrazIT) Study protocol

PONE-D-21-13311R1

Dear Dr. Camelo,

We’re pleased to inform you that your manuscript has been judged scientifically suitable for publication and will be formally accepted for publication once it meets all outstanding technical requirements.

Kind regards,

Dominique Heymann, Ph.D.

Academic Editor

PLOS ONE

---

## [Editor Report · Acceptance letter]

19 Aug 2021

PONE-D-21-13311R1 

Predictors of the outcome of immune tolerance induction in patients with haemophilia A and inhibitors: The Brazilian Immune Tolerance (BrazIT) Study protocol 

Dear Dr. Camelo:

I'm pleased to inform you that your manuscript has been deemed suitable for publication in PLOS ONE. Congratulations! Your manuscript is now with our production department. 

Kind regards, 

on behalf of

Pr. Dominique Heymann 

Academic Editor

PLOS ONE